# Learning to Trust Bellman Updates: Selective State-Adaptive Regularization for Offline RL

Qin-Wen Luo [* 1]   Ming-Kun Xie [* 1]   Ye-Wen Wang [1]   Sheng-Jun Huang [1]

## Abstract

Offline reinforcement learning (RL) aims to learn an effective policy from a static dataset. To alleviate extrapolation errors, existing studies often uniformly regularize the value function or policy updates across all states. However, due to substantial variations in data quality, the fixed regularization strength often leads to a dilemma: Weak regularization strength fails to address extrapolation errors and value overestimation, while strong regularization strength shifts policy learning toward behavior cloning, impeding potential performance enabled by Bellman updates. To address this issue, we propose the selective state-adaptive regularization method for offline RL. Specifically, we introduce state-adaptive regularization coefficients to trust state-level Bellman-driven results, while selectively applying regularization on high-quality actions, aiming to avoid performance degradation caused by tight constraints on low-quality actions. By establishing a connection between the representative value regularization method, CQL, and explicit policy constraint methods, we effectively extend selective state-adaptive regularization to these two mainstream offline RL approaches. Extensive experiments demonstrate that the proposed method significantly outperforms the state-of-the-art approaches in both offline and offline-to-online settings on the D4RL benchmark. The implementation is available at https://github.com/QinwenLuo/SSAR.

## 1. Introduction

Reinforcement Learning (RL) has made remarkable strides across diverse domains (Degrave et al., 2022; Kaufmann et al., 2023; Haarnoja et al., 2024). However, RL typically improves performance through trial and error, which limits its practical application to critical scenarios, such as healthcare decision-making (Tang & Wiens, 2021; Fatemi et al., 2022) and autonomous driving (Fang et al., 2022; Diehl et al., 2023). These tasks often have such strict requirements regarding potential risks and time costs that the interaction with the environment is limited or inaccessible.

Offline RL has mitigated this problem by deriving policies from static datasets without actual environment interactions. Direct application of the off-policy algorithms could result in value overestimation and extrapolation errors (Fujimoto et al., 2019; Kumar et al., 2019). To address these challenges, based on how they resolved around the policy iteration process, existing RL-based offline methods can be roughly divided into two groups, value regularization (Kumar et al., 2020; Cheng et al., 2022; Nakamoto et al., 2024) and policy constraint (Fujimoto & Gu, 2021; Kostrikov et al., 2021b; Nair et al., 2020). The former suppresses the Q-values of out-of-distribution (OOD) actions in the policy evaluation function to alleviate value overestimation; while the latter restricts updates near the dataset in the policy improvement function to avoid OOD actions. Both approaches incorporate regularization terms into their online counterparts, shifting the update from an optimistic to a pessimistic manner. Despite their demonstrated advances, existing methods still face the critical challenge of identifying optimal regularization strengths that balance trust in the outcomes derived from Bellman updates and the imitation of dataset actions to fully unlock the potential of RL-based learning.

Given that in model-free RL methods, performance gains beyond the behavior policy are fundamentally enabled by Bellman updates, it is crucial to balance confidence in these updates and conservative regularization. Since the balance is typically achieved by the fixed global coefficient in the regularization term, identifying its optimal value is paramount for effective performance. This presents challenges from three key aspects. Firstly, from a task-level perspective, the optimal global coefficient can vary significantly across different tasks. Certain tasks may benefit from a larger coefficient to enforce stricter regularization and avoid overestimation, while others may require a lower coefficient to enable more flexible policy updates, resulting in better performance

---

[*]Equal contribution [1]Nanjing University of Aeronautics and Astronautics, Nanjing, China. Correspondence to: Sheng-Jun Huang <huangsj@nuaa.edu.cn>.

*Proceedings of the $42^{st}$ International Conference on Machine Learning*, Vancouver, Canada. PMLR 267, 2025. Copyright 2025 by the author(s).

([Nakamoto et al., 2024](); [Lyu et al., 2022]()). Secondly, from the perspective of training dynamics, the optimal coefficient should evolve throughout the training process. When the policy is less reliable and poorly constrained by the dataset at the early stage, a larger coefficient is essential to restrict policy updates. As the training advances and the distribution of the learned policy becomes sufficiently close to the dataset, a lower coefficient may be more beneficial for states where the policy actions are close to the dataset, thanks to the reliable and reasonable generalization enabled by Bellman updates ([Beeson & Montana, 2022]()). Thirdly, considering the variability in data densities, it is intuitive to trust the Q-values learned by Bellman updates for states with high data density, while conversely tightening the constraints for states with low data density. Despite some nuances, existing methods predominantly determine the global coefficient through extensive experiments, which struggles to address these challenges effectively.

Another critical issue with the fixed global regularization lies in its impact on the efficiency of transitioning from offline-to-online (O2O) RL. The uniform regularization strength presents a dilemma: low regularization coefficients may amplify the risks of extrapolation errors through insufficient behavioral constraints, yielding poorly initialized policies. Conversely, overly high regularization coefficients induce significant discrepancies between the offline and online Q-values in value regularization methods, as well as misalignment between the learned policy and the policy directly induced by the critic in policy constraint methods. To enhance the efficiency of O2O RL, researchers have explored various approaches, including gradually relaxing constraint terms ([Beeson & Montana, 2022](); [Zhao et al., 2022]()) and adopting alternative action sampling strategies ([Zhang et al., 2023](); [Uchendu et al., 2023]()). However, these methods make modifications based on global coefficients, which prevents further improvement in fine-tuning efficiency.

In this work, we propose state-adaptive regularization that dynamically quantifies the reliability of Bellman updates, guiding the policy to trust optimistic outcomes at the state level. Instead of a fixed coefficient, our method employs a neural network as a state-dependent coefficient generator and adaptively updates it to adjust regularization strengths based on the discrepancy between the dataset and the distribution of the learned policy. To enhance its universality, we establish a connection between CQL, a representative value regularization method, and explicit policy constraint methods, extending it to both popular offline approaches. Furthermore, we propose a selective regularization method by selecting a subset of data with high rewards. This allows the policy update to focus on high-quality actions and avoid excessive constraints on low-quality parts. With the integration of selective state-adaptive regularization, we effectively bridge the gap between offline and online RL, achieving efficient O2O RL through simple linear coefficient annealing.

## 2. Preliminaries

**Offline RL** The environment in RL is typically modeled as a Markov Decision Process (MDP), which is defined by the tuple $(S, A, R, P, \mu, \gamma)$ ([Sutton & Barto, 2018]()), where $S$ is the state space, $A$ is the action space, $P : S \times A \to \Delta(S)$ is the transition function, $R : S \times A \to \mathbb{R}$ is the reward function, $\mu$ is the initial state distribution and $\gamma$ is a discount factor. The goal of RL is to find a policy $\pi : S \to \Delta(A)$ that maximizes the expected discounted return: $\mathbb{E}_{s_0 \sim \mu, a_t \sim \pi(\cdot|s_t), s_{t+1} \sim P(\cdot|s_t, a_t)} \left[ \sum_{t=0}^{\infty} \gamma^t R(s_t, a_t) \right]$. For any policy $\pi$, we define the state value function as $V^\pi(s) = \mathbb{E}_\pi[\sum_{t=0}^{\infty} \gamma^t R(s_t, a_t)|s_0 = s]$ and the state-action value function as $Q^\pi(s) = \mathbb{E}_\pi[\sum_{t=0}^{\infty} \gamma^t R(s_t, a_t)|s_0 = s, a_0 = \pi(\cdot|s_0)]$. The agent interacts with the environment by observing states, taking actions and receiving rewards, and improves the policy through the interactive data.

In offline RL, the agent has no access to the environment but only to a fixed dataset $D$, which is typically assumed to be collected by a behavior policy $\pi_\beta$. Existing works follow the key insight of maintaining pessimism in policy learning, which constrains the learned policy close to the dataset. For the value function and the policy, which are the two crucial components of model-free RL, value regularization and policy constraint are commonly applied in pursuit of this goal.

**Value regularization** In the policy iteration, OOD actions are inevitably taken to compute the Bellman targets for the $Q$ function update. This induces significant extrapolation error and overestimation of OOD actions ([Kumar et al., 2019](); [Fujimoto et al., 2019]()), ultimately resulting in poor or even negative performance improvement. Some work combats this problem by applying value regularization to the $Q$ function update to suppress the overestimation of OOD actions ([Cheng et al., 2022](); [Kumar et al., 2020](); [Nakamoto et al., 2024]()). A representative algorithm is conservative Q-learning (CQL) ([Kumar et al., 2020]()), which learns a conservative $Q$ function such that the expected values of a policy under this $Q$ function approximate the lower bound of its true values. The crucial modification is the value regularization applied to the $Q$ function update

$$\min_Q \beta \, \mathbb{E}_{s \sim D} \left( \log \sum_a \exp \left[ Q(s, a) \right] - \mathbb{E}_{a \sim D} \left[ Q(s, a) \right] \right)$$
$$+ \frac{1}{2} \mathbb{E}_{s, a, s' \sim D} \left[ \left( Q(s, a) - \hat{\mathcal{B}}^{\pi_k} \hat{Q}^k(s, a) \right)^2 \right]$$

$$(1)$$

The second term is the empirical Bellman error objective used to update the $Q$ function in RL, where the empiri-

cal Bellman operator is computed as $\hat{\mathcal{B}}^{\pi_k}\hat{Q}^k = r(s,a) + \gamma\mathbb{E}_{a'\sim\pi_k(\cdot|s')}[\hat{Q}^k(s',a')]$. The first term is a value regularizer, aiming to bound the $Q$ values to avoid overestimation of OOD actions, and $\beta$ is a tradeoff factor controlling the intensity of pessimism. Given that the regularization only affects the update of the Q function, the policy update follows the form of online RL. Akin to the implementation of SAC (Haarnoja et al., 2018), the policy of CQL is modeled as a Gaussian distribution but updates by approximating the Boltzmann distribution of $Q$ values with the following loss:

$$\min_{\pi} \mathbb{E}_{s\sim D, a\sim\pi(\cdot|s)}[\alpha\log\pi(a|s) - Q(s,a)] \quad (2)$$

**Explicit policy constraint**  Some other works focus on directly constraining the policy update without the change of $Q$ function update. TD3+BC (Fujimoto & Gu, 2021) is a minimalist but efficient approach by only adding a regularization term of the MSE loss between the actions output by the deterministic policy and the actions in the dataset. With the same $Q$ update function as TD3 (Fujimoto et al., 2018), the update loss functions are defined as

$$\min_{Q} \mathbb{E}_{s,a,s'\sim D}[(Q - [r(s,a) + \gamma\hat{Q}^k(s',\pi_k(s'))])^2] \quad (3)$$

$$\max_{\pi} \mathbb{E}_{s,a\sim D}[\lambda Q(s,\pi(s)) - (\pi(s) - a)^2] \quad (4)$$

where $\lambda = \alpha/\frac{1}{N}\sum_{(s_i,a_i)}|Q(s_i,a_i)|$ is an adaptive scalar that controls the intensity of the regularizer and normalizes the loss term about $Q$ values.

# 3. Selective State-Adaptive Regularization

In this section, to adaptively adjust the regularization strength across different states and maximize the potential benefits of Bellman updates, we propose the **Selective State-Adaptive Regularization** method. This approach comprises three key components: In Section 3.1, we establish the equivalence between Conservative Q-Learning (CQL) and explicit policy constraint methods. Building upon this relationship, we introduce a learnable state-adaptive regularization mechanism that dynamically adjusts state-wise regularization coefficients based on the divergence between the log-likelihood of dataset actions under the learned policy and a given threshold. In Section 3.2, we introduce a distribution-aware strategy based on the learned policy to automatically determine the appropriate threshold for this mechanism. To further ensure the validity of the constraints and fully exploit the advantages of the RL paradigm, we propose a selective regularization strategy in Section 3.3.

Additionally, in Section 3.4, we extend our method to deterministic policy algorithms that lack explicit policy distributions. Finally, Section 3.5 presents our online fine-tuning method, highlighting its simplicity and minimal reliance on the offline data.

## 3.1. State-Adaptive Coefficients

To address the challenges outlined in Section 1, we propose to use state-adaptive coefficients instead of a fixed global coefficient, which allows for adaptively controlling the strength of regularization at the state level. This modification exhibits two advantages. On the one hand, the adaptive update manner performs automatic adjustments to accommodate differences between tasks and dynamics across training stages; on the other hand, the state-adaptive nature enables us to determine the proper coefficients by leveraging Bellman-driven results, which are closely related to data density information.

Considering that different offline methods have different constraint objectives, it is challenging to directly apply state-adaptive coefficients to these methods. Recalling that the core principle behind these algorithms is to learn a desirable policy within a reliable region near the dataset, we provide a unified framework to integrate different constraint objectives.

We begin with a classic offline RL algorithm CQL (Kumar et al., 2020), which is also a representative of value regularization methods, and bridge its relationship with the explicit policy constraint methods.

**Proposition 3.1.** *With the policy $\pi$ modeled as a Boltzmann distribution, e.g. $\pi(a|s) \propto \exp(Q(s,a))$, the regularization term of Eq. (1) is equivalent to the negative log-likelihood term about $\pi$ at the dataset actions, that is*

$$\min_{Q} \beta\,\mathbb{E}_{s\sim D}\left[\log\sum_a \exp(Q(s,a)) - \mathbb{E}_{a\sim D}[Q(s,a)]\right]$$
$$\Updownarrow$$
$$\min_{\pi} \beta\,\mathbb{E}_{(s,a)\sim D}\left[-\log\pi(a|s)\right]$$

Proposition 3.1 shows that the equivalent regularization term can be considered as an explicit policy constraint, similar to SAC+ML used in (Yu & Zhang, 2023). This observation tells us that the coefficients in both approaches directly affect the probabilities of dataset actions in the learned policy. This motivates us to use a unified framework to adjust the coefficients for these two types of methods.

It becomes intuitive to relax the constraints when the policy actions are sufficiently close to the dataset, while enhancing them in regions of higher uncertainty. Building on this intuition, given the formulations of policy updates in Eq. (4) and Proposition 3.1, we propose a simple yet effective method to derive state-adaptive coefficients by constraining action probabilities to exceed the state-level thresholds determined by the policy distribution. Specifically, for a stochastic policy, we define the objective as

$$L_\beta(\phi) = \mathbb{E}_{(s,a)\sim D}\left[\log\pi(a|s) - C_n(s)\right]\beta_\phi(s) \quad (5)$$

where

$$C_n(s) = \min\{\log \pi(\mu + n\sigma|s), \log \pi(\mu - n\sigma|s)\} \quad (6)$$

where $\mu$ represents the mean of the learned policy, and $\sigma$ denotes its standard deviation. $\beta_\phi(s)$ is state-adaptive coefficients modeled by a neural network, and $n$ is a parameter controlling the width of the trust region. Considering that in some implementations of CQL, the policy is modeled as a squashed Gaussian distribution, we use the *min* operator to capture the appropriate thresholds.

With the proposed method, the state-adaptive coefficients will decrease when the probabilities of the dataset actions exceed the thresholds, and conversely, increase when they fall below them. This ensures that the learned policy not only restricts the updates within a reliable region but also obtains the potential performance improvements enabled by the generalization capabilities of Bellman updates.

By incorporating state-adaptive coefficients, we define the regularization term in CQL as

$$\min_Q \beta_\phi(s)\mathbb{E}_{s\sim D}[\log \sum_a \exp(Q(s,a)) - \mathbb{E}_{a\sim D}[Q(s,a)]]$$
$$+ \frac{1}{2}\mathbb{E}_{s,a,s'\sim D}[(Q - \hat{\mathcal{B}}^{\pi_k}\hat{Q}^k)^2]$$
$$(7)$$

which allows for a flexible trade-off between optimism and pessimism at the state level.

### 3.2. Distribution-Aware Thresholds

An important problem is how large the reliable region should be to impose an effective regularization. It is noteworthy that the size of a reliable region controlled by $n$ typically reflects the confidence of the learned policy based on Bellman updates. Due to the variability in task complexity, it is inherently challenging to determine a proper $n$ across different tasks. For instance, in the dataset with low data coverage, a larger value might be necessary to account for the increased uncertainty, whereas in well-sampled tasks, a lower one can avoid unnecessary constraints. Using a task-agnostic approach for determining $n$ is not a good choice and often yields unfavorable performance. This motivates the need for an adaptive mechanism that performs dynamical adjustments based on the characteristics of the task and the underlying policy distribution.

Intuitively, the parameter $n$ should be gradually increased throughout the training process. In the early stage, the policy distribution deviates significantly from the dataset, making it suffer from extrapolation errors. Once the policy is well-learned and sufficiently bounded around the dataset, looser constraints can allow the learning process to potentially benefit from Bellman updates. Given that the divergence

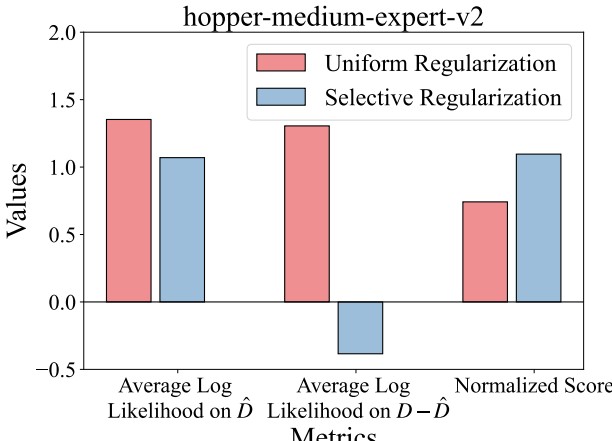

*Figure 1.* Comparison of Uniform vs. Selective Regularization. The values are evaluated by the policies trained by with different regularization on all data in the dataset.

between the policy distribution and the dataset can reflect the dynamic of the training progress, we propose to perform an adjustment of $n$ in a distribution-aware manner to dynamically expand the trust region.

We start by setting the initial value of $n$ to a low value $n_{start}$, ensuring sufficient constraint, and gradually increasing $n$ until it reaches $n_{end}$. Throughout the training process, a simple linear schedule for $n$ is applied to relax the constraint:

$$n =\leftarrow n + \Delta n \quad (8)$$

where $\Delta n = (n_{end} - n_{start}) \cdot T_{inc}/T$, $T_{inc}$ represents the update interval and $T$ represents the total steps.

By using this approach, the threshold $n$ is initially set to a small value, causing the loss in Eq. (5) to be negative. As a result, the coefficients of most states will increase during the early stages to enforce the policy distribution towards the dataset. As training progresses, the learned policy gradually approximates the dataset actions across most states. With the expansion of the trust region, the constraints on certain states are gradually relaxed, allowing for broader exploration. To reach a balance in the strength of the constraint, we terminate the update once the condition $\mathbb{E}_{(s,a)\sim D}[\log \pi(a|s) - C_n(s)] > 0$ is satisfied. This adaptive scheme ensures that the trust region evolves in alignment with the distribution of the policy.

### 3.3. Selective Regularization

Although we can adaptively control the strength of regularization with state-adaptive coefficients, it has still proven to be difficult to obtain favorable performance on low-quality datasets. As shown in Figure 1, this is because the proposed coefficients updating mechanism encourages the policy to

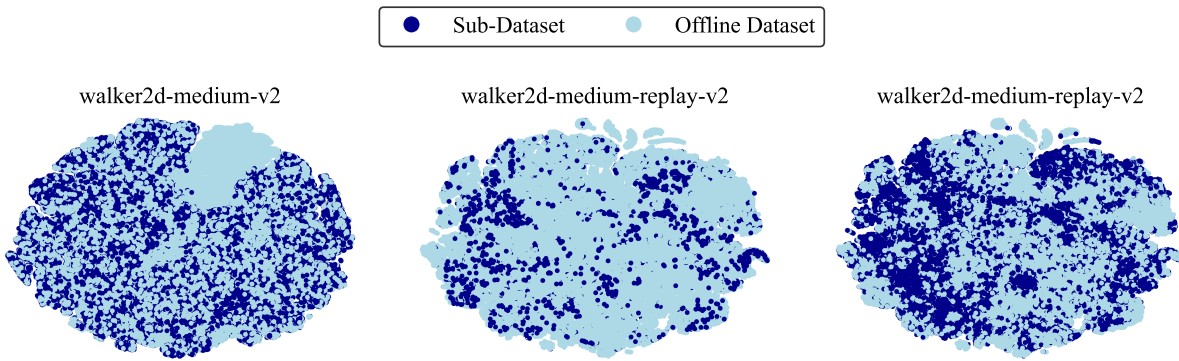

(a) Data with high returns in the dataset with a low quality variance
(b) Data with high returns in the dataset with a high quality variance
(c) Data with positive advantages in the dataset with a high quality variance

*Figure 2.* t-sne visualization of different sub-dataset selection methods

output high probabilities for the actions within the dataset, including those that lead to suboptimal or even poor performance. To address this problem, we propose a selective regularization strategy to impose the constraint on a subset of high-quality actions, which prevents the policy from being misguided by low-quality actions. To achieve this, we prioritize trajectories with high returns, emphasizing the regularization on valuable actions, while overlooking other constraints with minimal returns.

we construct a sub-dataset denoted by $\hat{D}$ by selecting trajectories with returns greater than a selective threshold $G_T$. To mitigate the negative impact of low-quality actions, we update the state-adaptive coefficients only within $\hat{D}$ as

$$L_\beta(\phi) = \mathbb{E}_{(s,a)\sim\hat{D}}[\log \pi(a|s) - C_n(s)]\beta_\phi(s) \quad (9)$$

where $C_n(s)$ is still computed as Eq. (6).

When the sub-dataset $\hat{D}$ can cover the most region of the whole dataset $D$, we perform regularization selectively on the actions with the sub-dataset. For CQL, the objective can be reformulated as

$$\min_Q \beta_\phi(s)\mathbb{E}_{s\sim\hat{D}}[\log\sum_a \exp(Q(s,a)) - \mathbb{E}_{a\sim D}[Q(s,a)]]$$
$$+ \frac{1}{2}\mathbb{E}_{s,a,s'\sim D}[(Q - \hat{\mathcal{B}}^{\pi_k}\hat{Q}^k)^2] \quad (10)$$

Generally, there exist two situations for different datasets. For datasets with low quality variances, the sub-dataset consisting of high-return trajectories can effectively capture the distribution of the offline dataset, as shown in Figure 2(a). This motivates us to use selective regularization for increasing the Q-values of valuable actions in the dataset while allowing the Q-values of low-quality actions to update naturally through the Bellman backup. This mitigates the overestimation of Q values by leveraging the generalization of similar actions or states to constrain the policy.

For the datasets composed of heterogeneous data with a wide-ranging distribution, such as *walker2d-medium-replay-v2*, which records all samples in the replay buffer during training, the sub-dataset can only cover a portion of the entire distribution, as shown in Figure 2(b). In such a situation, solely adjusting constraints on the states within the sub-dataset may cause catastrophic overestimation for the remaining states.

To address this issue, we utilize the approaches from IQL Kostrikov et al. (2021b) to pre-train a state-action value network $Q$ and a state value network $V$ to capture the relative value of the data. As a representative offline RL algorithm, IQL has demonstrated the superiority of advantage-weighted behavior cloning, where Q-values and V-values are learned through expectile regression in a non-iterative manner with in-sample learning

$$L_V = \mathbb{E}_{(s,a)\sim D}[L_2^\tau(Q(s,a) - V(s))]$$
$$L_Q = \mathbb{E}_{(s,a,s')\sim D}[(r(s,a) + \gamma V(s') - Q(s,a))^2] \quad (11)$$

where $L_2^\tau(x) = |\tau - \mathbb{I}(x < 0)|x^2$ and $\mathbb{I}(\cdot)$ is the indicator function.

With the pre-trained $Q$ and $V$, we can filter the valuable actions on the whole dataset by the condition $Q(s,a) - V(s) > 0$. Analogous to Eq. (9) and Eq. (10), we can construct the sub-dataset $\hat{D}$ by the data that satisfies the metric and favor these selected actions. Given that this approach guarantees that at least one action is constrained for all states, the sub-dataset can capture the approximate distribution of the offline dataset, as shown in Figure 2(c).

### 3.4. Extend to Explicit Policy Constraint Methods

Based on the relationship between the value regularization and the explicit policy constraint methods as shown in Proposition 3.1, the idea of the selective regularization method can also be applied to explicit policy constraint methods. In this

section, we provide an extension of the proposed method to the representative algorithm TD3+BC. However, it is infeasible to directly update the state-adaptive coefficients using Eq. (9), as TD3+BC models the policy as a deterministic form. Recalling that the interaction process in TD3, we find that the exploration noise behaves similarly to the standard deviation, allowing us to treat the actions as samples from a Gaussian distribution with the mean of the policy actions and a fixed standard deviation. This modification enables us to rewrite Eq. (9) as

$$L_\beta(\phi) = \mathbb{E}_{(s,a)\sim\hat{D}}[n^2\delta^2 - (a - \pi(s))^2]\beta_\phi(s) \quad (12)$$

where $\delta$ is the exploration noise in TD3, which is usually set as 0.1. Note that Eq. (9) and Eq. (12) are grounded in the same underlying mechanism, with the primary difference lying in the specific policy formulations to which they are applied. See Appendix C for the derivation.

By incorporating the state-adaptive coefficients, we define the objective of the policy improvement in TBC+BC as

$$\max_\pi \mathbb{E}_{s,a\sim D}[Q_{norm}(s, \pi(s)) - \beta_\phi(s)(\pi(s) - a)^2] \quad (13)$$

Similarly, for the datasets with low variance in quality, the policy can be only constrained near the sub-dataset $\hat{D}$

$$\max_\pi \mathbb{E}_{s,a\sim D}[Q_{norm}(s, \pi(s)) \\ - \mathbb{I}\left((s,a)\in\hat{D}\right)\beta_\phi(s)(\pi(s) - a)^2] \quad (14)$$

where $\mathbb{I}\left((s,a)\in\hat{D}\right)$ denotes an indicator function that specifies whether the data point $(s,a)$ belongs to the sub-dataset $\hat{D}$, $Q_{norm}(s, \pi(s))$ follows the normalization trick of TD3+BC (Fujimoto & Gu, 2021) and can be computed as $Q(s, \pi(s))/\frac{1}{N}\sum_{(s_i,a_i)}|Q(s_i,a_i)|$.

### 3.5. Efficient Offline-to-Online RL

Generally, the state-adaptive coefficients can be viewed as the confidence of the policy in a given state. When the coefficient is low, the policy update closely follows the results of bellman updates. Similar to the online approaches, this update has the optimistic property that encourages the policy to explore higher-value regions. When the coefficient is high, it serves to constrain the policy near the dataset for safety, as the critic updated by the Bellman backup may guide the policy toward unreliable regions. These state-adaptive coefficients allow the policy to determine whether to trust the critic at the state level, enabling a flexible trade-off between optimism and pessimism, which efficiently minimizes the discrepancies between the offline and online update mechanisms.

Additionally, since the policy is trained on an offline dataset that spans a wide range of states, the coefficients can be directly applied during the online fine-tuning stage. Two factors ensure stable performance improvements: i) there are far fewer out-of-distribution (OOD) actions during the interaction phase, and ii) the coefficient network can generalize its adaptability to new states.

Thanks to the advantages of state-adaptive coefficients, we can efficiently implement online fine-tuning by fixing the parameters of the coefficient network, with linear annealing applied to the outputs as

$$\beta_{on}(s) = \min\{1 - \frac{N}{N_{end}}, 0\}\cdot\beta(s) \quad (15)$$

where $N$ is the number of online interaction steps and $N_{end}$ is the given number of decay steps.

During online fine-tuning, the update functions of the actor and the critic hold on and $\beta_{on}(s)$ is used to replace the regularization coefficients.

Although the form of our method is similar to unified approaches across O2O phases, such as IQL, our method offers advantages due to the well-trained coefficient network and its generalization. As a result, we can reserve only a subset of trajectories for initializing the online replay buffer, or even discard the offline dataset entirely and use only the online data to update the policy. This is infeasible for the unified methods, as they require the offline dataset to constrain the policy update, thereby hindering the utilization efficiency of online data and resulting privacy breaches in some scenarios. On the other hand, with low coefficients in some states, the Q-values are not severely underestimated, which is helpful for avoiding drastic jumps and the consequent performance degradation for value regularization methods. For explicit policy constraint methods, low coefficients enable the actor to update in the direction of the maximal Q-values, which prevents potentially inaccurate evaluations.

Since the offline policy is well-trained to collect online data with higher quality, we apply the constraint term for all online data to achieve stable performance improvement.

The complete training procedure is outlined in the pseudo-code provided in Algorithm 1.

## 4. Experiments

In this section, we incorporated the proposed method with CQL (Kumar et al., 2020) and TD3+BC (Fujimoto & Gu, 2021) and conducted extensive experiments to validate the effectiveness on D4RL (Fu et al., 2020) MuJoCo and AntMaze tasks, including HalfCheetah, Hopper, Walker2d and AntMaze environments. In Section 4.1, we first demonstrate significant performance improvements compared to the backbone methods, CQL and TD3+BC. Then, in Section 4.2, we highlight the effectiveness of our approach in

*Table 1.* Offline performance comparison with backbone algorithms TD3+BC (Fujimoto & Gu, 2021) and CQL (Kumar et al., 2020) on the D4RL benchmark. We evaluate the D4RL normalized scores of standard base algorithms (denoted as "Base") against those augmented with selective state-adaptive regularization (referred to as "Ours"). The symbol $\pm$ represents the standard deviation across the seeds. Superior scores are highlighted in bold.

| Dataset | TD3+BC | | CQL | | Avg. | |
|---|---|---|---|---|---|---|
| | Base | Ours | Base | Ours | Base | Ours |
| halfcheetah-m-v2 | 48.3±0.2 | **56.5±3.7** | 47.1±0.2 | **63.9±1.2** | 47.7 | **60.0** |
| hopper-m-v2 | 58.7±3.9 | **101.6±0.4** | 65.6±3.5 | **89.1±9.7** | 62.1 | **95.4** |
| walker2d-m-v2 | 82.3±2.2 | **87.9±2.4** | 81.6±1.2 | **84.9±1.7** | 81.9 | **86.4** |
| halfcheetah-mr-v2 | 44.4±0.6 | **49.6±0.3** | 45.7±0.4 | **53.8±0.4** | 45.0 | **51.7** |
| hopper-mr-v2 | 66.4±27.1 | **101.6±0.7** | 92.3±9.3 | **101.4±2.1** | 79.3 | **101.5** |
| walker2d-mr-v2 | 81.6±7.1 | **93.5±2.0** | 79.2±1.9 | **94.7±3.3** | 80.4 | **94.1** |
| halfcheetah-me-v2 | 92.9±2.0 | **94.9±1.2** | 93.0±4.2 | **102.1±1.2** | 93.0 | **98.5** |
| hopper-me-v2 | 101.4±8.2 | **103.8±6.7** | 97.8±8.6 | **109.6±3.2** | 99.6 | **106.7** |
| walker2d-me-v2 | 110.3±0.5 | **112.5±1.4** | 109.2±0.2 | **112.2±0.9** | 109.8 | **112.4** |
| halfcheetah-e-v2 | **95.9±1.1** | 95.5±1.3 | 97.0±0.5 | **105.9±0.9** | 96.5 | **100.7** |
| hopper-e-v2 | 108.4±3.6 | **109.8±4.3** | 108.7±2.8 | **111.4±0.2** | 108.6 | **110.6** |
| walker2d-e-v2 | **110.1±0.5** | 109.6±0.3 | 110.1±0.2 | **110.2±0.2** | **110.1** | 110.0 |
| **locomotion total** | 1000.8 | **1116.7** | 1030.4 | **1139.1** | 1015.6 | **1128.0** |
| **95% CIs** | 917.9~1083.7 | 1096.2~1137.3 | 990.4~1070.1 | 1111~1167.3 | 937.5~1078.6 | 1093.2~1162.8 |
| umaze-v2 | 88.6±4.6 | **93.4±3.3** | 92.8±1.5 | **96.0±2.3** | 90.7 | **94.7** |
| umaze-diverse-v2 | 43.2±18.8 | **50.0±5.4** | 27.8±13.1 | **80.2±7.9** | 35.5 | **65.1** |
| medium-play-v2 | 0.0±0.0 | **49.4±3.4** | 67.0±4.2 | **70.2±6.7** | 33.5 | **59.8** |
| medium-diverse-v2 | 0.0±0.0 | **47.6±12.1** | 60.5±9.2 | **71.6±9.3** | 30.3 | **59.6** |
| large-play-v2 | 0.0±0.0 | **18.0±4.6** | 24.8±9.8 | **53.0±4.1** | 12.4 | **35.5** |
| large-diverse-v2 | 0.0±0.0 | **17.6±9.8** | 21.2±12.1 | **35.8±18.9** | 10.6 | **26.7** |
| **antmaze total** | 131.8 | **276.0** | 294.1 | **406.8** | 213.0 | **341.4** |
| **95% CIs** | 78.2~185.5 | 246.5~305.5 | 230.9~357.3 | 334.9~478.7 | 130.1~295.8 | 273.7~409.1 |

the offline-to-online setting by applying a simple annealing scheme to the state-adaptive coefficients. Lastly, to validate the contributions of individual components, we perform ablation studies to evaluate the impact of the state-adaptive coefficients and the selection methods for the sub-dataset $\hat{D}$. All methods are run with four random seeds, and the averaged results are reported.

### 4.1. Performance in Offline Setting

We first demonstrate a significant performance improvement over the backbone methods, CQL and TD3+BC, as shown in Table 1. Our method consistently outperforms both CQL and TD3+BC across a wide range of datasets, with particularly notable improvements in datasets containing lower-quality data, such as *hopper-medium-v2*. We

attribute this improvement primarily to the unique ability of our method to selectively identify and focus on valuable sub-datasets, combined with the enhanced generalization enabled by bellman updates with state-adaptive coefficients.

By leveraging the sub-dataset $\hat{D}$, the policy is selectively constrained to focus on high-value actions, resulting in conservative updates but with high returns. Additionally, the state-adaptive coefficients allow Bellman updates to generalize more effectively, achieving superior performance within a reliable region near the sub-dataset.

In a manner similar to TD3+BC, we incorporate a behavior cloning term $\mathbb{E}_{(s,a)\sim D, a'\sim\pi(\cdot|s)} (a'-a)^2/2$ into the policy update for CQL in the Antmaze tasks to ensure stable performance. We also compare our method with several advanced offline methods in Appendix B.1, further highlighting the

Table 2. Comparison of online fine-tuning (250k steps) performance with the baseline algorithms on D4RL benchmark.

| Dataset | IQL | SPOT | FamCQL | CQL | TD3+BC | TD3+BC(SA) | CQL(SA) |
|---|---|---|---|---|---|---|---|
| halfcheetah-medium-v2 | 49.7 | 58.6 | 65.3 | 48.0 | 52.5 | 82.9±2.5 | **95.3±1.5** |
| hopper-medium-v2 | 75.2 | 99.9 | 101.0 | 63.8 | 63.7 | **103.5±0.4** | 99.3±3.8 |
| walker2d-medium-v2 | 80.8 | 82.5 | 93.3 | 82.8 | 86.6 | 101.6±7.4 | **105.9±3.7** |
| halfcheetah-medium-replay-v2 | 45.2 | 57.6 | 73.1 | 49.4 | 49.3 | 73.1±3.0 | **79.4±2.3** |
| hopper-medium-replay-v2 | 91.1 | 97.3 | 102.8 | 101.3 | 97.0 | 102.9±0.9 | **103.1±0.2** |
| walker2d-medium-replay-v2 | 89.2 | 86.4 | 103.6 | 87.9 | 89.9 | 100.9±5.4 | **116.3±2.1** |
| halfcheetah-medium-expert-v2 | 92.4 | 91.9 | 95.7 | 95.7 | 93.2 | 98.5±4.1 | **115.4±1.5** |
| hopper-medium-expert-v2 | 109.6 | 106.5 | 104.4 | 110.8 | 99.8 | **111.2±2.9** | 109.5±5.4 |
| walker2d-medium-expert-v2 | 115.0 | 110.6 | 110.4 | 109.8 | 115.8 | 115.7±5.5 | **117.5±2.5** |
| halfcheetah-expert-v2 | 96.4 | 94.1 | 106.5 | 97.3 | 95.8 | 102.5±0.9 | **113.3±0.8** |
| hopper-expert-v2 | 100.3 | 111.8 | 109.6 | 111.9 | 109.5 | **112.0±2.4** | 110.8±1.6 |
| walker2d-expert-v2 | 112.5 | 109.9 | 112.6 | 109.7 | 111.4 | **113.8±0.5** | 112.6±1.2 |
| **locomotion total** | 1057.4 | 1107.1 | 1178.3 | 1068.4 | 1064.5 | 1218.6 | **1278.4** |
| **95% CIs min** | 981.5 | 1093.1 | 1165.3 | 1058.9 | 1039.8 | 1165.4 | 1254.9 |
| **95% CIs max** | 1133.2 | 1121.4 | 1191.5 | 1080.1 | 1089.2 | 1248.9 | 1303.5 |
| antmaze-umaze-v2 | 83.0 | 98.8 | - | 95.2 | 72.8 | 96.5±3.2 | **99.0±0.6** |
| antmaze-umaze-diverse-v2 | 38.2 | 56.8 | - | 59.2 | 39.8 | 87.2±5.0 | **95.0±2.5** |
| antmaze-medium-play-v2 | 78.8 | **92.5** | - | 77.0 | 0.0 | 76.5±16.3 | 88.0±2.4 |
| antmaze-medium-diverse-v2 | 80.2 | 87.0 | - | 84.0 | 0.2 | 63.0±36.1 | **89.0±3.2** |
| antmaze-large-play-v2 | 42.8 | 60.0 | - | 51.8 | 0.0 | 35.5±13.2 | **66.5±13.4** |
| antmaze-large-diverse-v2 | 40.2 | **63.0** | - | 38.2 | 0.0 | 30.5±15.0 | 56.8±18.2 |
| **antmaze total** | 363.2 | 458.1 | - | 405.5 | 112.8 | 389.2 | **494.3** |
| **95% CIs min** | 302.8 | 384.0 | - | 327.9 | 79.5 | 316.1 | 415.6 |
| **95% CIs max** | 423.7 | 534.0 | - | 483.1 | 146.1 | 462.4 | 573.4 |

superior performance of our approach.

## 4.2. Performance in Offline-to-Online Setting

We conducted experiments in the O2O setting with 250,000 online steps and compared the results with several unified O2O methods, including CQL, TD3+BC, and advanced approaches such as IQL (Kostrikov et al., 2021b), SPOT (Wu et al., 2022), and FamO2O (Wang et al., 2024). FamO2O employs hierarchical models to determine state-adaptive improvement-constraint balances. For our experiments, we used an implementation of FamO2O integrated with CQL, referred to as FamCQL. However, since FamCQL did not provide its hyperparameters for the AntMaze tasks, we tested the default settings but observed poor performance. As a result, its scores are not included in the comparisons.

Table 2 illustrates the competitive performance of our

method in O2O tasks, even with a simple linear annealing of the coefficients, where (SA) indicates the use of our approach. We attribute this to the fact that adaptive constraints in the offline process can reduce the difference between offline and online updates, thereby ensuring effective exploration in online fine-tuning.

Another advantage of our approach is that it can eliminate the need for access to the offline dataset during online fine-tuning, as discussed in Section 4.2. For better performance improvements, in our experiments, we initialized the online replay buffer with a subset of high-return trajectories during online fine-tuning for Mujoco tasks, but with the entire offline dataset for Antmaze tasks. We also conducted experiments with other initialization strategies and listed the results in Appendix B.2. The results demonstrate that our method achieves competitive fine-tuning performance even without access to the offline dataset or when utilizing only a

subset of it.

### 4.3. Ablation Study

In this subsection, we conduct ablation studies on the effectiveness of the state-adaptive coefficients and the selective regularization, particularly the IQL-style selection method for datasets with high quality variance.

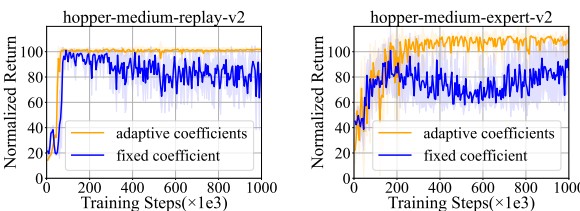

*Figure 3.* Offline performance comparisons of different types of the coefficient used for the regularization.

Figure 3 shows the necessity of the state-adaptive coefficients, where the fixed coefficient is applied globally. As discussed in Section 1, the global coefficient constrains the policy update to a narrow region of the sub-dataset, limiting the ability to achieve performance beyond it. In contrast, the state-adaptive coefficients allow policy updates within a reliable region induced by bellman updates, enabling performance that exceeds the dataset.

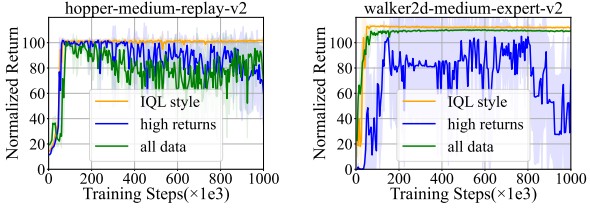

*Figure 4.* Offline performance comparisons of different selection methods for the sub-dataset.

From Figure 4, for datasets with a wide-ranging distribution, the sub-dataset composed solely of high-return trajectories can lead to significant performance degradation and volatility. This happens because the sub-dataset represents only a portion of the overall distribution and may be overly narrow, while the entire dataset is used for updating Q-values. As a result, the policy update can drift toward unreliable regions in out-of-distribution states, which subsequently affects updates for other states. The sub-dataset extracted using the IQL-style approach can mitigate this issue, as data is selected at the state level rather than the trajectory level, resulting in a broader yet equally valuable distribution.

### 5. Conclusions

To unlock the potential performance improvements enabled by Bellman updates and address the challenges posed by cross-task, cross-training stage, and varying data densities in uniform regularization, we propose state-adaptive regularization coefficients on selective states. We replace the global coefficient with state-adaptive coefficients and adaptively adjust them based on the policy distribution by the relationship between CQL and explicit policy methods. Additionally, we pre-select sub-datasets containing high-quality actions to reasonably relax the constraints. Empirically, extensive experiments show that our approach offers a significant improvement over various baseline methods, achieving state-of-the-art performance on the D4RL benchmark.

### Acknowledgements

This work was supported by the NSFC (U2441285, 62222605) and the Natural Science Foundation of Jiangsu Province of China (BK20222012).

### Impact Statement

This paper presents work whose goal is to advance the field of Machine Learning. There are many potential societal consequences of our work, none which we feel must be specifically highlighted here.

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

# A. Related Work

**Offline RL**    In offline RL, inaccessible interactions with the environment pose challenges such as extrapolation error and overestimation due to out-of-distribution (OOD) actions (Fujimoto et al., 2019; Kumar et al., 2019). Current model-free offline RL approaches can be categorized into two main groups based on their constraint forms. Value regularization methods learn a conservative value function to mitigate overestimation (Cheng et al., 2022; Kumar et al., 2020; Bai et al., 2022; Ma et al., 2021; Kostrikov et al., 2021a), while policy constraint methods explicitly or implicitly constrain policy updates close to the behavior policy (Fujimoto & Gu, 2021; Wu et al., 2022; Wang et al., 2022b; Kostrikov et al., 2021b; Nair et al., 2020). Recent methods have added regularization terms to the policy objective, deriving the regularization term for both the updates of policy and Q-values (Garg et al., 2023; Xu et al., 2023; Tarasov et al., 2024). Some model-based offline RL methods achieve exceptional performance (Yu et al., 2020; 2021; Sun et al., 2023; Luo et al., 2023; Zhu et al., 2025) due to the generalization capabilities of learned models. Additionally, recent works have framed policy learning as a supervised learning and sequence modeling problem using advanced network architectures (Emmons et al., 2021; Chen et al., 2021; Janner et al., 2022). However, due to the pessimistic constraints for reliable learning, several hyperparameters need to be determined before employing offline algorithms, most of which are empirical (Paine et al., 2020). Furthermore, most current algorithms use a uniform constraint coefficient across all states, despite variations in data density.

**Value regularization**    Value regularization methods penalize the Q-values of OOD actions, thereby reducing the probability of selecting such actions as the policy is derived from the Q-values. Some works address extrapolation error and overestimation by applying regularization constraints to suppress the Q-values of OOD actions. CQL (Kumar et al., 2020) augments the standard Bellman error objective with a Q-value regularizer that provides a lower bound on the value of the current policy. Various CQL variants adjust the constraints or modify the regularizer to avoid excessive pessimism (Lyu et al., 2022; Nakamoto et al., 2024; Mao et al., 2024; Yu et al., 2021). Ensemble-based approaches (An et al., 2021; Nikulin et al., 2022) use a similar Bellman target by estimating uncertainty to mitigate the overestimation of OOD actions. Notably, one study demonstrated the connection between value regularization and one-step RL (Eysenbach et al., 2023) by modifying the entire policy evaluation formula, while we reveal the link between the regularizer of CQL and explicit policy constraint methods.

**Policy constraint**    Unlike value regularization methods, policy constraint methods maintain the policy evaluation formula but constrain policy updates during the improvement stage. TD3+BC (Fujimoto & Gu, 2021) illustrates the effectiveness of a simple policy constraint applied to the off-policy TD3 algorithm (Fujimoto et al., 2018). Recent works focus on explicit policy constraints for stochastic policies (Wu et al., 2022; Nair et al., 2020) or more expressive policies (Kang et al., 2024; Wang et al., 2022a). Another implementation of policy constraints is the implicit policy constraint derived from the KL-divergence between the learned policy and the behavior policy, known as advantage-weighted behavior cloning (Peng et al., 2019). Many works based on this approach have demonstrated significant performance improvements across various tasks (Nair et al., 2020; Kostrikov et al., 2021b; Wang et al., 2024; Park et al., 2024; Hansen-Estruch et al., 2023). Among them, FamO2O (Wang et al., 2024) also utilizes state-adaptive coefficients, but it is limited to IQL-style methods with implicit constraints and the entire dataset. Moreover, the coefficients are updated by maximizing Q-values, lacking the interpretability offered by our method.

**Offline-to-online RL**    The distribution shift between the offline dataset and that induced by the learned policy is the most challenging issue in offline-to-online RL. Policy constraint methods can mitigate this distribution shift by keeping policy updates close to the replay buffer. For transitioning from offline RL to online settings, some methods adopt the key idea of policy constraints while adjusting constraints for improved online performance, such as decaying the constraint coefficient in a certain way (Beeson & Montana, 2022; Zhao et al., 2022; Wang et al., 2024). Additionally, certain offline methods can be directly applied to O2O RL (Nair et al., 2020; Kostrikov et al., 2021b). However, the use of a uniform constraint coefficient makes it difficult to balance stability and speed, inhibiting efficient online fine-tuning. Conversely, transitioning from value regularization offline methods to online poses a challenge due to the sudden increase in Q-values, resulting from the pessimistic estimation of OOD actions in offline RL. During online fine-tuning, the agent will estimate the Q-values of OOD actions optimistically, leading to a rapid increase in overall Q-values and subsequent performance degradation. Some works have relaxed constraints to prevent severe underestimation of OOD actions during offline learning (Lyu et al., 2022; Nakamoto et al., 2024) or utilized Q-ensemble methods to maintain reliable estimations at a higher computational cost (Lee et al., 2022; Mark et al., 2022; Zhao et al., 2023). In contrast, our method assigns different levels of pessimism based on the distance between the output of the learned policy and the reliable sub-dataset, avoiding unreasonable underestimation. Thus,

*Table 3.* Offline performance comparison with prior methods on the D4RL benchmark. The mean-wise best results among algorithms are highlighted in bold.

| Dataset | AWAC | IQL | Cal-QL | SPOT | FamCQL | TD3+BC(SA) | CQL(SA) |
|---|---|---|---|---|---|---|---|
| halfcheetah-m | 49.8±0.3 | 48.1±0.3 | 47.8±0.2 | 57.6±0.6 | 58.1±0.5 | 56.5±3.7 | **63.9±1.2** |
| hopper-m | 68.6±11.2 | 66.7±4.4 | 64.7±3.4 | 71.4±37.2 | 82.3±16.0 | **101.6±0.4** | 89.1±9.7 |
| walker2d-m | 85.1±0.5 | 74.8±1.8 | 84.3±0.9 | 69.6±30.2 | 87.4±0.6 | **87.9±2.4** | 84.9±1.7 |
| halfcheetah-m-r | 45.4±0.6 | 44.5±0.3 | 46.2±0.3 | 52.3±0.7 | 51.9±1.3 | 49.6±0.3 | **53.8±0.4** |
| hopper-m-r | 97.8±1.4 | 89.6±11.9 | 93.4±6.6 | 87.1±23.9 | 85.4±14.2 | **101.6±0.7** | 101.4±2.1 |
| walker2d-m-r | 73.2±8.4 | 80.6±5.8 | 84.7±1.4 | 88.9±1.6 | 88.6±1.5 | 93.5±2.0 | **94.7±3.3** |
| halfcheetah-m-e | 95.3±0.9 | 91.8±2.1 | 52.7±5.4 | 92.7±2.3 | 91.7±2.8 | 94.9±1.2 | **102.1±1.2** |
| hopper-m-e | 108.6±2.3 | 106.3±7.4 | 107.6±2.4 | 102.1±8.8 | 92.9±14.8 | 103.8±6.7 | **109.6±3.2** |
| walker2d-m-e | 89.7±39.4 | 111.9±1.0 | 109.0±0.3 | 110.3±0.2 | 110.6±0.4 | **112.5±1.4** | 112.2±0.9 |
| halfcheetah-e | 97.1±0.5 | 95.1±3.1 | 96.1±0.9 | 94.4±0.5 | 93.4±0.9 | 95.5±1.3 | **105.9±0.9** |
| hopper-e | 99.7±11.1 | 111.1±2.2 | **111.9±0.3** | 110.1±3.9 | 111.7±1.3 | 109.8±4.3 | 111.4±0.2 |
| walker2d-e | 112.7±0.3 | **113.0±0.2** | 109.1±0.3 | 110.1±0.2 | 110.1±0.3 | 109.6±0.3 | 110.2±0.2 |
| **locomotion total** | 1023 | 1033.5 | 1007.5 | 1046.6 | 1064.1 | 1116.7 | **1139.1** |
| antmaze-u | 63.5±19.2 | 74.8±5.8 | 74.8±1.8 | 88.8±3.3 | - | 93.4±3.3 | **96.0±2.3** |
| antmaze-u-d | 57.8±8.0 | 52.2±6.4 | 16.2±20.1 | 41.5±5.3 | - | 50.0±5.4 | **80.2±7.9** |
| antmaze-m-p | 0.0±0.0 | 63.8±4.8 | 69.5±8.0 | 63.0±13.9 | - | 49.4±3.4 | **70.2±6.7** |
| antmaze-m-d | 0.0±0.0 | 61.2±2.5 | 64.0±5.6 | 67.0±21.5 | - | 47.6±12.1 | **71.6±9.3** |
| antmaze-l-p | 0.0±0.0 | 37.8±3.8 | 41.8±4.5 | 34.0±7.1 | - | 18.0±4.6 | **53.0±4.1** |
| antmaze-l-d | 0.0±0.0 | 20.8±6.1 | 32.8±9.3 | 36.2±11.6 | - | 17.6±9.8 | **35.8±18.9** |
| **antmaze total** | 121.3 | 310.6 | 299.1 | 330.5 | - | 286 | **406.8** |

the Q-values estimated during the offline stage will not fall significantly below the values evaluated during the online stage, preventing drastic jumps in Q-values. Furthermore, the adaptive coefficients can relax the constraints in some states and achieve more effective fine-tuning.

## B. More Experimental Results

### B.1. Comparison with some advanced offline methods

We also compared our method with several common advanced baseline methods. The results shown in Table 3 demonstrate the competitive performance of our method.

### B.2. Different initializations for online replay buffer

A notable advantage of our method is its ability to reduce reliance on offline datasets, as the state-adaptive coefficients can generalize to OOD data encountered during the online fine-tuning process. We conducted O2O experiments by initializing the online replay buffer with four different types: (1) Initialize the buffer with the entire offline dataset akin to (Kostrikov et al., 2021b). (2) Conduct a separate online buffer and sample symmetrically from both the offline dataset and online buffer akin to (Ball et al., 2023). (3) Initialize the buffer with the sub-dataset $\hat{D}$ used in offline training. (4) Initialize the buffer without any offline data. For simplicity, we denote them as *all*, *half*, *part* and *none* respectively.

From the results shown in Table 4, we can see that in non-sparse reward tasks, our method achieves good fine-tuning performance even without accessing the offline dataset. In challenging sparse reward tasks, fine-tuning with offline data performs better due to the difficulty in accurately capturing the value function distribution (especially for TD3+BC). In our

implementation, we initialized the online replay buffer during fine-tuning with a high-value subset of the offline dataset for Mujoco tasks, whereas for Antmaze tasks, the entire offline dataset was used.

*Table 4.* The performance comparison of online fine-tuning with different sample strategies.

| Dataset | TD3+BC(SA) | | | | CQL(SA) | | | |
|---|---|---|---|---|---|---|---|---|
| | *all* | *half* | *part* | *none* | *all* | *half* | *part* | *none* |
| halfcheetah-medium-v2 | 61.2 | 63.4 | 82.9 | 66.6 | 71.2 | 70.6 | 95.3 | 69.8 |
| hopper-medium-v2 | 102.6 | 103.1 | 103.5 | 97.0 | 102.1 | 77.7 | 99.3 | 89.4 |
| walker2d-medium-v2 | 99.6 | 101.5 | 101.6 | 101.8 | 86.6 | 105.3 | 105.9 | 104.1 |
| halfcheetah-medium-replay-v2 | 65.9 | 69.4 | 73.1 | 71.6 | 59.9 | 61.8 | 79.4 | 62.4 |
| hopper-medium-replay-v2 | 100.7 | 101.4 | 102.9 | 100.9 | 103.2 | 103.0 | 103.1 | 102.8 |
| walker2d-medium-replay-v2 | 96.2 | 98.0 | 100.9 | 101.6 | 101.5 | 110.2 | 116.3 | 105.9 |
| halfcheetah-medium-expert-v2 | 99.0 | 98.0 | 98.5 | 99.9 | 117.8 | 103.3 | 115.4 | 106.1 |
| hopper-medium-expert-v2 | 98.1 | 111.9 | 111.2 | 87.4 | 110.8 | 111.3 | 109.5 | 110.3 |
| walker2d-medium-expert-v2 | 116.4 | 113.1 | 115.7 | 109.7 | 114.9 | 112.4 | 117.5 | 113.8 |
| halfcheetah-expert-v2 | 103.4 | 102.5 | 102.5 | 103.3 | 109.1 | 108.5 | 113.3 | 104.7 |
| hopper-expert-v2 | 107.6 | 108.8 | 112.0 | 98.9 | 110.4 | 110.8 | 110.8 | 100.6 |
| walker2d-expert-v2 | 110.6 | 111.4 | 113.8 | 112.6 | 111.7 | 111.6 | 112.6 | 112.1 |
| **locomotion total** | 1154.8 | 1180.9 | 1218.6 | 1165.0 | 1196.0 | 1192.1 | 1279.4 | 1190.1 |
| antmaze-umaze-v2 | 96.5 | 95.5 | 95.8 | 95.8 | 99.0 | 99.2 | 97.8 | 97.8 |
| antmaze-umaze-diverse-v2 | 87.2 | 61.5 | 78.8 | 83.5 | 95.2 | 93.8 | 95.0 | 95.0 |
| antmaze-medium-play-v2 | 76.5 | 59.0 | 18.8 | 2.8 | 88.8 | 84.2 | 86.2 | 83.5 |
| antmaze-medium-diverse-v2 | 63.0 | 38.0 | 38.8 | 2.0 | 89.0 | 89.8 | 88.0 | 88.5 |
| antmaze-large-play-v2 | 35.5 | 31.2 | 14.0 | 5.8 | 66.5 | 57.8 | 58.2 | 53.0 |
| antmaze-large-diverse-v2 | 30.5 | 35.2 | 18.2 | 9.2 | 56.8 | 42.5 | 26.5 | 26.5 |
| **antmaze total** | 389.2 | 320.4 | 264.4 | 199.1 | 495.3 | 467.3 | 451.7 | 444.3 |

### B.3. Computational cost

Our method introduces a lightweight network with minimal overhead. Below is the training time on a 2080 GPU, excluding IQL-style pretraining, showing a slight increase in time cost.

| Algorithm | TD3+BC | TD3+BC(SA) | CQL | CQL(SA) |
|---|---|---|---|---|
| Time Cost(h) | 2.25 | 2.69 | 7.22 | 8.05 |

*Table 5.* The computational time required by TD3+BC, CQL, and the integration of our proposed method.

## C. Proof

**Derivation of Eq.** (12)   Revisiting the interaction process in TD3, when the policy takes an action, the exploration noise is sampled from a Gaussian distribution $N(0, \delta)$ and added to the output $\pi(s)$ of the policy. Thus, the action can be viewed as a sample from a stochastic policy $N(\pi(s), \delta)$, denoted as $\pi_{act}$.

We can explicitly compute the logarithmic value of the policy $\pi_{act}$ as follows

$$\log \pi_{act}(a|s) = -\log \sqrt{2\pi}\delta - \frac{(a - \pi(s))^2}{2\delta^2} \tag{16}$$

By plugging it to Eq. (9), we can get

$$
\begin{aligned}
L_\beta(\phi) &= \mathbb{E}_{(s,a)\sim\hat{D}}[\log \pi_{act}(a|s) - C_n(s)]\beta_\phi(s) \\
&= \mathbb{E}_{(s,a)\sim\hat{D}}[-\log \sqrt{2\pi}\delta - \frac{(a - \pi(s))^2}{2\delta^2} + \log \sqrt{2\pi}\delta + \frac{n^2\delta^2}{2\delta^2}]\beta_\phi(s) \\
&= \frac{1}{2\delta^2}\mathbb{E}_{(s,a)\sim\hat{D}}[n^2\delta^2 - (a - \pi(s))^2]\beta_\phi(s)
\end{aligned}
\tag{17}
$$

where $C_n(s) = \min\{\log \pi_{act}(\pi(s) + n\delta|s), \log \pi_{act}(\pi(s) - n\delta|s))\} = \log \pi_{act}(\pi(s) \pm n\delta|s)$.

Omit the coefficient $1/2\delta^2$, we can get Eq. (12).

**Proof of Proposition 3.1**  Revisit the original formula of CQL($\rho$) (Eq. (3) in (Kumar et al., 2020) with the regularizer of KL-divergence)

$$\min_Q \max_\mu \beta(\mathbb{E}_{s\sim D, a\sim\mu(a|s)}[Q(s,a)] - \mathbb{E}_{s,a\sim D}) + \frac{1}{2}\mathbb{E}_{s,a,s'\sim D}[(Q(s,a) - \hat{\mathcal{B}}^{\pi_k}\hat{Q}^k(s,a))^2] - D_{KL}(\mu||\rho) \tag{18}$$

Consider the inner optimization problem for a more general form

$$\max_\mu \beta(\mathbb{E}_{x\sim\mu}[f(x)] - D_{KL}(\mu||\rho) \text{ s.t. } \sum_x \mu(x) = 1, \mu(x) \geq 0 \ \forall x \tag{19}$$

The optimal solution is equal to

$$\mu^*(x) = \frac{\rho(x)\exp(f(x))}{\sum_x \rho(x)\exp(f(x))} \tag{20}$$

If we set $f(x) = Q(s,a)/\alpha$, then we can get

$$\min_Q \underbrace{\beta \cdot \alpha \cdot \mathbb{E}_{s\sim D}[\log \sum_a \exp(Q(s,a)/\alpha) - \frac{1}{\alpha}\mathbb{E}_{a\sim D}[Q(s,a)]]}_{\text{regularizer } R} + \frac{1}{2}\mathbb{E}_{s,a,s'\sim D}[(Q - \hat{\mathcal{B}}^{\pi_k}\hat{Q}^k)^2] \tag{21}$$

As the policy $\pi$ is modeled as a Boltzmann distribution, we can get

$$\forall(s,a) \in S \times A, \ \pi(a|s) = \frac{\exp(Q(s,a))/\alpha}{\sum_a \exp(Q(s,a))/\alpha} \tag{22}$$

Then

$$\forall(s,a) \in S \times A, \ \log \sum_a \exp(Q(s,a)/\alpha) = \frac{1}{\alpha}Q(s,a) - \log \pi(a|s) \tag{23}$$

Select actions that follow the distribution of the behavior policy, and plug Eq. (23) into the regularizer $R$ in Eq. (21)

$$
\begin{aligned}
R &= \beta \cdot \alpha \cdot \mathbb{E}_{s\sim D}[\mathbb{E}_{a\sim D}[\frac{1}{\alpha}Q(s,a) - \log \pi(a|s)] - \frac{1}{\alpha}\mathbb{E}_{a\sim D}[Q(s,a)]] \\
&= \beta\mathbb{E}_{s,a\sim D}[-\log \pi(a|s)]
\end{aligned}
\tag{24}
$$

# D. pseudo code

---

**Algorithm 1** state-adaptive Regularization for offline and offline-to-online RL

---

**Input:** the return threshold $G_T$, the initial value $n_{start}$, the end value $n_{end}$, update interval $T_{inc}$, offline total steps $T$, online decay steps $N_{end}$

Initialize Q network $Q$, V network $V$ with random parameters for IQL-style update

Initialize critic network $Q_\psi$, actor network $\pi_\omega$, coefficient network $\beta_\phi$ with random parameters for offline policy learning, replay buffer $D$ with the offline dataset

 # *Pre-training*

Compute the returns $G$ for all trajectories or train $Q$ and $V$ by Eq. (11)

Obtain the sub-dataset $\hat{D}$ by $G > G_T$ or $Q(s, a) - V(s) > 0$

 # *Offline training*

**for** iteration $i = 1, 2, \cdots, T$ **do**

 Sample a mini-batch $B = (s, a, r, s', d)$ from $D$, where $d$ is the done flag

 Update $\pi_\omega$ by Eq. (2) (Eq. (13) or Eq. (14) for TD3+BC)

 Update $Q_\psi$ by Eq. (7) or Eq. (10) (Eq. (3) for TD3+BC)

 Update $\beta_\phi$ by Eq. (9) (Eq. (12) for TD3+BC)

 **if** $i \% T_{inc} == 0$ **then**

  **if** $\mathbb{E}_{(s,a)\sim D}\left[\log \pi(a|s) - C_n(s)\right] \leq 0$ **then**

   Increase $n$ by Eq. (8)

  **else**

   Terminate the update of $n$

  **end if**

 **end if**

**end for**

 # *Online training*

Reset the replay buffer $D$ according to the given strategy

**for** iteration $i = 1, 2, \cdots$ **do**

 Interact with the environment and store the new transition in the replay buffer $D$

 Obtain the annealed coefficients $\beta_{on}$ by Eq. (15)

 Update $\pi_\omega$ by Eq. (2) (Eq. (13) for TD3+BC)

 Update $Q_\psi$ by Eq. (7) (Eq. (3) for TD3+BC)

**end for**

---

# E. Implementation details

## E.1. Baseline implementation

We reproduce all results of baseline algorithms, except for FamCQL, using the deep offline RL library CORL (Tarasov et al., 2022) (https://github.com/tinkoff-ai/CORL). For FamCQL, we reproduce the results according to the official code (https://github.com/LeapLabTHU/FamO2O). All hyperparameters are kept consistent with the official implementations.

## E.2. Network structure of the coefficient generator

We employ a three-layer MLP architecture with 512 neurons per hidden layer to parameterize $\beta(s)$. The sigmoid activation function is adopted in the final output layer, with output values constrained to the interval $(0, 1.5\beta_{init})$, where $\beta_{init}$ denotes the regularization coefficient from the original implementations of both CQL and TD3+BC algorithms.

Additionally, to prevent catastrophic value overestimation, we applied the trick of Layernorm on the critic networks.

## E.3. Return thresholds

Since the critic induced by the deterministic policy is less robust, the return thresholds of TD3+BC are slightly lower than that of CQL. The return thresholds are shown in Table 6. For the dataset involving *replay* data, we use the IQL method to filter valuable actions. The value of $\tau$ is set as 0.7, the same as the original implementation of IQL.

Furthermore, in Antmaze tasks, substantial fluctuations were observed in the number of valuable actions acquired through

IQL pre-training. Therefore, we construct the sub-dataset $D$ by selecting trajectories that complete the task successfully.

*Table 6.* The return thresholds for different tasks.

| Dataset | CQL(SA) | TD3+BC(SA) |
|---|---|---|
| halfcheetah-medium-v2 | 6000 | 5200 |
| hopper-medium-v2 | 2500 | 1800 |
| walker2d-medium-v2 | 3600 | 2500 |
| halfcheetah-expert-v2 | 11000 | 10500 |
| hopper-expert-v2 | 3500 | 3500 |
| walker2d-expert-v2 | 4800 | 4500 |

### E.4. Distribution-aware threshold setting

For the distribution-aware threshold, in Mujoco tasks, we set $n_{start}$ to 1 and $n_{end}$ primarily to 3 (1.5 for expert datasets to better mimic high-quality actions). In Antmaze tasks, we set $n_{end}$ to 5 for more efficient exploration. Other details about the hyperparameters can be found in our open-source code.

### E.5. Offline-to-online setting

After offline training, the policy interacts with the environment for 5,000 warm-up steps without any parameter updates. The decay steps $N_{end}$ is set to 400,000, while the total number of interaction steps in our experiments is limited to 250,000. For hard Antmaze tasks in TD3+BC(SA), we keep the coefficients unchanged during the online fine-tuning because the coefficients are relatively small after the offline process. And we set the interval for policy updates to a larger value to achieve stable updates.

