# OpenReview forum: "Learning to Trust Bellman Updates: Selective State-Adaptive Regularization for Offline RL"
_ICML.cc/2025/Conference — ICML 2025 poster_

### Official Review · Reviewer_578t · 2025-03-07

**Overall Recommendation:** 3

**Summary:**

This paper proposes selective state-adaptive regularization method for offline RL, addressing the limitations of fixed-strength regularization. It involves state-adaptive regularization coefficients through learning and regularization on a subset of high-quality data. Experiments on the D4RL benchmark show some performance improvements in both offline and offline-to-online scenarios.

**Claims And Evidence:**

The claims are well-supported by clear and convincing evidence.

**Essential References Not Discussed:**

Yes. The references contain the background required to understand the key contributions of the paper. No essential literature has been omitted in the paper citations, and relevant research findings/results/etc. have been cited and discussed.

**Experimental Designs Or Analyses:**

The experimental designs are clear and the analyses are complete. Through a variety of comparisons and ablation experiments, it demonstrates the performance of the proposed method in different scenarios. However, the experimental part lacks sensitivity analysis of hyperparameters.

**Methods And Evaluation Criteria:**

The proposed state-adaptive coefficient and selective regularization are simple yet effective. The use of D4RL benchmarks aligns standard settings in RL.

**Other Comments Or Suggestions:**

There are problems with the expression, for example, "we provide a unified framework to unified framework..."

**Other Strengths And Weaknesses:**

Strengths:

1) The proposed selective state-adaptive regularization method is simple yet effective.
2) Sufficient theoretical derivations are provided.

Weaknesses:

1) The comparative results are not sufficiently reliable as lacking the performance of some latest offline and offline-to online methods. There is no mention of hyperparameter sensitivity and setting experiment.
2) Current explanations regarding threshold calculation are confusing and hard to follow, requiring further clarification.

**Questions For Authors:**

1) What does $\mu$ represent in Eq.6?
2) Why not fine-tune regularization coefficients during online training? Would adaptive updates improve stability?
3) Does updating regularization coefficients introduce significant computational overhead?
4) Could the authors compare the proposed method with FamO2O in detail, especially in terms of interpretability?

**Relation To Broader Scientific Literature:**

This paper addresses the limitations of prior fixed-strength regularization works. It includes the comparison results with recent state-adaptive approaches like FamO2O.

**Theoretical Claims:**

The overall derivation is logically rigorous and coherently structured, and the proof of Proposition 3.1 is theoretically correct.

---

> ### Author Rebuttal · Authors · 2025-04-01
>
> Thanks for your thorough review and positive recognition of our work. We are glad that you consider our work "well-supported, simple yet effective, logically rigorous and coherently structured." We are glad to answer all your questions.
>
> **Q1:More comparative results**
>
> **A1:** We compare our approach with two strong baselines, MCQ and MISA. MCQ is a CQL variant achieving mild conservatism, while MISA unifies CQL and TD3+BC via mutual information. The results  below demonstrates our competitive performance.
> |Dataset |MCQ|MISA|TD3+BC(SA) |CQL(SA)
> |:---:|:---:|:---:|:---:|:---:|
> |ha-m|62.7|47.4|56.5|**63.9**|
> |ho-m|79.2|67.1|**101.6**|89.1|
> |wa-m|**90.5**|84.1|87.9|84.9|
> |ha-m-r|**56.6** |45.6|49.6|53.8|
> |ho-m-r|100.7|98.6|**101.6**|101.4|
> |wa-m-r|91.4| 86.2|93.5|**94.7**|
> |ha-m-e|88.2| 94.7|94.9|**102.1**|
> |ho-m-e|**110.6**|109.8|103.8|109.6|
> |wa-m-e|**114.0**|109.4|112.5|112.2|
> |ha-e|96.6|95.9|95.5|**105.9**|
> |ho-e|110.9|111.9|109.8|**111.4**|
> |wa-e|107.4|109.3|109.6|**110.2**|
> |total|1108.8|1060|1116.7|**1139.1**|
>
> Since we do not employ any specialized techniques in the online phase, we compare our approach with some other advanced unified methods across O2O phases, as shown below. Our method still leads in performance.
> |Dataset |SPOT|Cal-QL|TD3+BC(SA) |CQL(SA)
> |:---:|:---:|:---:|:---:|:---:|
> |ha-m|58.6|61.6|82.9|**95.3**|
> |ho-m|99.9|97.1|**103.5**|99.3|
> |wa-m|82.5|83.4|101.6|**105.9**|
> |ha-m-r|57.6|51.1|73.1|**79.4**|
> |ho-m-r|97.3|99.3|102.9|**103.1**|
> |wa-m-r|86.4|91.9|100.9|**116.3**|
> |ha-m-e|91.9| 95.9|98.5|**115.4**|
> |ho-m-e|106.5| **111.4**|111.2|109.5|
> |wa-m-e|110.6|110.9|115.7|**117.5**|
> |ha-e|94.1|97.0|102.5|**113.3**|
> |ho-e|111.8|**112.2**|112.0|110.8|
> |wa-e|109.9|109.5|**113.8**|112.6|
> |total|1108.8|1121.3|1218.6|**1278.4**|
>
> **Q2: Experiments on hyperparameter sensitivity**
>
> **A2:** We conduct hyperparameter sensitivity experiments on  $n_{end}$​, as shown below. The results demonstrate the robustness of our method, as the early stopping mechanism in the update of $n$ ensures that $n$ increases to an appropriate value and then stabilizes.
> |$n_{end}$ |1.0|2.0|3.0|4.0|5.0|
>  :---:|:---:|:---:|:---:|:---:|:---:|
> |ho-m |82.6±26.7|79.4±25.6|89.1±9.7|90.6±9.7|84.1±10.5|
> |ho-m-r |101.7±1.4|101.3±1.8|101.4±2.1|100.0±3.1|97.1±9.4|
>
> **Q3:The meaning of $\mu$ in Eq.(6)**
>
> **A3:** Apologies for the omission. $\mu$ represents the mean of the stochastic policy (i.e., the learned policy), and $\sigma$ denotes its standard deviation.
>
> **Q4:Explanation regarding threshold**
>
> **A4:** The threshold $n$ in Eq.(5) is initially set to a small value (1.0 in our experiments), which results in a large $C_n(s)$. At this stage, the loss value of Eq.(5) is negative, leading the regularization coefficient to increase and encouraging the learned policy to closely imitate dataset actions during the early learning phase. As $n$ gradually increases, the loss value of Eq.(5) eventually becomes positive, indicating that the learned policy has sufficiently approximated the high-value dataset actions in most states. In this trust region, certain state constraints can be relaxed while ensuring that a substantial portion of constraints remain enforced.
>
> **Q5:Fine-tuning coefficients online**
>
> **A5:** Since a single online step cannot directly determine action value, directly fine-tuning regularization online is difficult.
>
> **Q6:Computational cost**
>
> **A6:** Our method introduces a lightweight network with minimal overhead. Below is the training time on a 2080 GPU, excluding IQL-style pretraining, showing a slight increase in time cost.
> |Algo|TD3+BC|CQL|TD3+BC(SA) |CQL(SA)|
> |:---:|:---:|:---:|:---:|:---:|
> |Time Cost(h)|2.25|2.69|7.22|8.05|
>
> **Q6:Comparison with FamO2O**
>
> **A6:** Both methods adjust state-wise regularization, but key differences include:
> 1. Regularization Parameterization: FamO2O employs a hierarchical architecture, using an intermediate variable to learn policies under different constraints. In contrast, our method directly parameterizes the regularization coefficients with a neural network, explicitly modeling constraint strength across different states.
> 2. Policy Representation: FamO2O learns a family of policies, where the optimal policy is selected by maximizing the Q-value with respect to the intermediate variable. This requires passing through two networks to obtain the final policy. In contrast, our method updates the constraints based on the relationship between the learned policy and dataset actions, directly learning the final policy without additional selection steps.
> 3. Interpretability: The intermediate variable in FamO2O lacks interpretability, meaning its value does not directly indicate constraint strength. In our approach, the parameterized regularization coefficients directly reflect the constraint strength, providing better interpretability.
>
> Thank you for your insightful review again. We hope we have resolved your concerns. We are always willing to answer any of your further concerns.

---

### Official Review · Reviewer_wb2q · 2025-03-12

**Overall Recommendation:** 3

**Summary:**

This paper introduces an offline RL method that balances regularization strength conditioned on the provided state. This state-adaptive form of regularization is applied to CQL and explicit policy constraint methods demonstrating improved performance in both offline and offline-to-online settings.

**Claims And Evidence:**

It was difficult to clearly articulate what claims this paper proposes as they are not stated anywhere in the paper. The paper claims to unify value and policy regularization approaches using state-adaptive regularization but still resorts to developing separate methods for each class of offline RL algorithm. This was fairly disappointing. Additionally there was a lot of narrative strength given to creating a wholly adaptive approach yet I was disappointed to find a lot of hand-tuned or scheduled parameterizations of the underlying objectives.

**Essential References Not Discussed:**

I felt that the paper fairly covered the offline RL literature. I would however recommend that the authors consider `Moskovitz, et al (NeurIPS 2021) “Tactical optimism and pessimism for deep RL”` which presents an approach to dynamically select between optimism and pessimism based on the task presented to the algorithm. I think it would help anchor the current work despite not being wholly aligned to the offline setting.

**Experimental Designs Or Analyses:**

Yes, I felt that the experiments were well set up. There is however a lot of analysis to support the claims of the scale and direction of state-adaptive regularization that should have been included. I am reduced my final score as a result of this.

**Methods And Evaluation Criteria:**

The paper attempts to unify the two major forms of regularization among contemporary offline RL approaches by bridging between value regularization and policy constraint methods. This appears to be begun by proposition 3.1 but the result of the proposition goes unused in the development of the objective set out in Equations 6 and 7.

I am generally supportive of creating state-adaptive approaches to dealing with the inherent partial observability problem. However, this paper seems to shift the “global” parametrization of regularization to the selection of the hyperparameter $n$ which defines the shape of the trust region. Even with a linear annealing of $n$ it’s not totally satisfying that so much attention was paid to set up prior regularization methods as inflexible or limited when there is a similar (yet downplayed) assumption being made in this work. The distributionally aware notion of the threshold is nice in principle but it rests on a pretty strong a priori assumption (line 185-188). As the threshold is updated on a set schedule with an early stopping condition, I am not comfortable calling this an adaptive scheme. As implemented and discussed in this paper, this threshold is exactly task/state/data distribution-agnostic. This same complaint exists for the selective threshold $G_T$ introduced in Section 3.3 as well as the annealing of the regularization coefficient in online fine-tuning of the offline policy (equation 15).

There also doesn’t seem to be a whole lot of consistency about what the specified objectives are for the proposed approach. There is fairly little organization around what objectives are used and when and how they all relate together. There appears to be a CQL version of the state-adpative regularization as well as a TD3+BC version, both with pretrained value functions from IQL? With different formulations depending on the quality of the dataset (e.g. differences between Eqts 13 and 14?)

There is a lot of speculative lanugage about the scale of the regularization coefficient reflecting confidence in the quality of the dataset. It would be nice to have this demonstrated with actual analyses rather than speculative language. This is continued when talking about the advantages of the proposed approach over IQL (lines 308-314). It would be nice to have these assumptions clearly identified because the advantage of the proposed approach really rests on the quality of the trained regularization coeffcient network. Assuming it’s accurate and generalizes to the online setting is great but perhaps is wishful thinking? This is further exacerbated by the claim at the end of Section 3 (lines 326-329); there is no evidence for this, only willful speculation.

**Other Comments Or Suggestions:**

It seems that Equation 5 is misreferenced in the paragraph before the equation is introduced?

It’s unclear where the quantities $\mu$ and $\sigma$ are drawn from in the definition of Equation 6. The term “trust region” is used but the specific computation of this distributional region is not specified.

I think that Section E in the appendix should be referenced far more prominently in the paper. It isn’t super clear but it does help unify the various techniques used in the paper.

## After rebuttal and reviewer discussion periods

I apologize for the lack of engagement the authors rightfully deserved from this submission. I believe that the authors did a nice job responding to the various requests made in the collective reviews. As such, I feel that all reviewers agreed that this is a paper worthy of acceptance, I did not feel inclined to increase my score after the author's rebuttal and in respect to the other reviews (and accompanying rebuttals). I urge the authors to certainly include all of their promised changes in the event that the paper is formally published.

**Other Strengths And Weaknesses:**

## Strengths

The conceptual framing of the work is well grounded in limitations of current offline RL methods. I’m not overly convinced of the originality of the ideas but it’s clear that the authors have attempted to unify regularization techniques and have thought through various challenges in doing so by adapting several approaches to combine in forming their proposed method.

I also commend the authors for their efforts to develop a method that works in standard offline RL as well as in the offline-to-online setting.

The empirical results presented in Section 4 are compelling and cover a wide range of problem complexity and distributional settings (here, discussing the formation of the offline dataset).

## Weaknesses

It’s unclear how the different objectives combine as the separate learnable parametrizations between the policy and state-adaptive regularization coefficient are not consistently parameterizied.

There are a lot of moving pieces and definition of objectives to establish the proposed state-adaptive regularization. As the paper is currently written, it is difficult to follow everything. There are a lot of “stream of consciousness” declarations which belies a poorly composed work. Specifically, so much of the paper has been talking about CQL up until late in Section 3.3 where suddenly “we utiliize the approaches from IQL” crops up. The paper would greatly benefit from a top down restructuring around what the proposed contributions are and how they’re acheived. The lack of clarify and construction of the paper also led me to lower my score of this paper.

**Questions For Authors:**

The analysis around Figure 1 is unclear. What determines the values of the histograms? What data is used? How early in the training is this analysis drawn from? I found the concept aroud Section 3.3 interesting but the presentation and writing is quite poor. Figure 1 should probably not be referenced until more detail is shared about what is being presented. (Porbably after Equation 10). Ultimately, there is not enough detail in place to describe what’s happening in Figure 1 to fully follow what is being presented and the ultimate value of the analysis.

What dictates the choice of $G_T$ in Section 3.3? Is this task-dependent?

Ultimately Section 3.3 is extremely unclear… Is there a two-phase training paradigm? First for the high-value sub-dataset and then a more general training over all data?

Is the $n$ in Equation 12 the same $n$ used to define the trust region?

Is policy learning only done on the sub-datasets?

**Relation To Broader Scientific Literature:**

I feel that the paper is well oriented among the relevant offline RL literature. There is fair discussion of prior work and its limitations as well as extensive baseline comparisons (mostly included in the appendix).

**Theoretical Claims:**

Not closely. Ultimately, I felt that the paper eschewed its theoretical claims pretty early on and simply compiled various components from prior literature when forming its own proposed method.

---

> ### Author Rebuttal · Authors · 2025-04-01
>
> Thanks for your insightful review and positive recognition of our paper. We appreciate the questions you raised and are committed to delivering a comprehensive response to address the issues.
>
> **Q1: Questions about claims**
>
> **A1:** Our main claim is derived from Proposition 3.1 and can be succinctly stated as balancing pessimism and optimism by constraining the divergence between the likelihood of high-quality dataset actions under the learned policy and a predefined threshold. This motivates the development of our state-adaptive mechanism, which adjusts regularization dynamically as divergence varies across different actions. Since the learned policy's distribution is directly used to update the regularization coefficients, we propose distinct methods tailored for stochastic and deterministic policies. However, the core insight remains the same for both, as detailed in the derivation of Eq. (12) in Appendix C. Additionally, due to the variability in dataset distributions and differences in reward scales, some manual hyperparameter tuning is unavoidable.
>
> **Q2: Questions about methods**
>
> **A2:** 1.) For the use of Proposition 3.1, as discussed in A1, it reveals that the regularizer in CQL inherently increases the likelihood of dataset actions under the learned policy. Building on this insight, we formulated Eq. (5) to strike a balance between pessimism and optimism.
> 2.) For $G_T$ and the schedule used to update $n$, a pre-defined metric is essential to select  high-value actions. In contrast, the schedule is not strictly necessary. If training time is sufficient, $n$ can be updated based on some metrics, such as the loss. Such pre-defined hyperparameters are common in RL, and the adaptivity in our approach is primarily reflected in the **state-dependent regularization coefficients** across varying states.
> 3.) For the different objectives in our approach, as shown in Fig. 2, a sub-dataset of high-return trajectories may not cover most of the offline dataset, leading to over-optimism in uncovered regions. To address this, we pre-train IQL critics (independent of the learned policy) to filter a reliable high-value sub-dataset that covers most of the offline dataset.
> 4.) Lastly, for the "speculative lanugage", our claims are empirically supported by the ablation study and the experimental results presented in Table 4.
>
> **Q3: Weakness 1 about the separate learnable parametrizations**
>
> **A3:** As discussed in A1, Eq. (5) applies to the stochastic policy, while Eq. (12) is for the deterministic policy (since the deterministic policy lacks an explicit policy distribution); however, both share the same key insight.
>
> **Q4: Weakness 2 about construction of the paper**
>
> **A4:** We introduced the technique of IQL to address the sub-dataset selection problem, which is independent of CQL.
> Regarding the paper structure, our goal is to make adaptive adjustments to the constraints across different states in order to maximize the potential benefits of Bellman updates. We begin by proposing a state-level regularization updating mechanism in Section 3.1, stemmed from Proposition 3.1. To automatically select the appropriate threshold for this mechanism, we introduce a method based on distributed perception in Section 3.2. To further ensure that constraints are valid and to fully harness the benifit of the RL form, we propose selective regularization in Section 3.3. Additionally, we extend the method to deterministic policy algorithms that lack explicit policy distributions.
> We will provide a summary at the end of each sub-section and emphasize that Eq. (5) and Eq. (12) are fundamentally the same but apply to different policy formulations. Thank you for your constructive suggestions.
>
> **Q5: The definition of $\mu$ and $\sigma$ in Eq.(6)**
>
> **A5:** $\mu$ is the mean of a stochastic policy and $\sigma$ is the standard deviation.
>
> **Q6: Details of Fig.1**
>
> **A6:** We trained the policy both with and without selective regularization, and then used the trained policy along with all data to compute the values presented in Fig. 1. In Fig. 1, we highlight the advantage of selective regularization: it helps to avoid the imitation of low-value actions.
>
> **Q7: Selection of $G_T$**
>
> **A7:** The selection of $G_T$ depends on the dataset, similar to previous works like DT and RvS, as reward scales vary significantly. Since the dataset's reward information is available, it's easy to determine a proper value.
>
> **Q8: Two-phase training paradigm**
>
> **A8:** Yes, we first pre-select the high-value sub-dataset before initiating offline training.
>
> **Q9: $n$ used  in Eq.(12)**
>
> **A9:** Yes, details are in Appendix C.
>
> **Q10: The region of policy learning**
>
> **A10:** The policy is updated across all states, but regularization related to the policy (if applied) is restricted to the sub-dataset.
>
> Thanks for your review again. We hope our response sufficiently addresses your concerns, and we remain available for any further clarifications.

---

### Official Review · Reviewer_ofHf · 2025-03-13

**Overall Recommendation:** 3

**Summary:**

The paper introduces a selective state-adaptive regularization method for offline RL to address the challenge of extrapolation errors caused by varying data quality. Unlike existing methods that apply uniform regularization across all states, the proposed approach learns adaptive regularization coefficients and selectively applies regularization only to high-quality actions, preventing performance degradation from over-constraining low-quality data. Extensive experiments on the D4RL benchmark show that this approach significantly outperforms state-of-the-art offline and offline-to-online RL methods.

**Claims And Evidence:**

The main claims seem reasonable and supported by evidence.

**Essential References Not Discussed:**

No  that I am aware of.

**Experimental Designs Or Analyses:**

Seems reasonable.

**Methods And Evaluation Criteria:**

The methods seem good for this type of work. The experiments are done on many standard RL test environments, and the baseline algorithms seem reasonably chosen. The only problem, unless I missed something, the code has not been provided. Moreover, there is lacking information on how the offline data has been collected, this seems important given the focus of the paper.

**Other Comments Or Suggestions:**

None

**Other Strengths And Weaknesses:**

One of the main weaknesses of the paper is that its contribution appears relatively incremental. While the idea of more refined regularization is useful, it feels like a minor extension of existing work, which limits its overall impact. Additionally, the paper lacks theoretical contributions, unlike related work that provides convergence guarantees, sample complexity bounds, or similar results. It would be beneficial to establish similar theoretical results to strengthen the paper’s contribution. Alternatively, providing simple illustrative examples where the benefits of the proposed method are clearly demonstrated could help justify its significance.

On the positive side, the paper is well-written and presents a methodologically sound approach. The evaluation is rigorous, and the results show clear improvements over baseline algorithms. These empirical gains support the practical value of the method, even if the theoretical foundation could be stronger.

**Questions For Authors:**

None

**Relation To Broader Scientific Literature:**

Seems reasonable.

**Theoretical Claims:**

There is just one theoretical statement, Proposition 1. It seems quite straight forward characterization of the regularization term, similar as is done in similar papers, e.g., (Kumar et al., 2020). I check the main steps in the proof, it is short and easy to follow, I think it is correct. One small feedback to the author is that the text in Proposition 1 could be improved, maybe split into two sentences, I felt it was difficult to read.

---

> ### Author Rebuttal · Authors · 2025-04-01
>
> Thank you for your thorough review and positive recognition of our work. We appreciate your thoughtful feedback and are pleased that you found our paper to be "well-written, methodologically sound, and of practical value." We also appreciate the questions you raised and are committed to delivering a comprehensive response to address your concerns in detail.
>
> **Q1: Code provision**
>
> **A1:** We have carefully verified that the code has been included in the supplementary material.
>
> **Q2: The collection of offline data**
>
> **A2:** Our approach is evaluated on the publicly available D4RL benchmark [1], which provides diverse offline datasets. These datasets are collected using various strategies, including hand-designed controllers, human demonstrators, multi-task agents performing different tasks in the same environment, and policy mixtures. Given that prior works on offline RL have typically used these datasets without explicitly describing their collection process, we followed the same convention. Further details regarding dataset collection can be found in the original D4RL paper.
>
> **Q3: Relatively incremental contribution and lack of theoretical contribution**
>
> **A3:** Existing approaches employ global regularization to constrain policy updates across the entire dataset. However, this can lead to excessive pessimism, limiting the agent's ability to harness the full benefits of Bellman updates. Building on the core insight of pessimism in offline RL (Proposition 3.1), we propose a selective state-adaptive regularization mechanism that dynamically adjusts regularization coefficients based on the likelihood of high-quality dataset actions under the learned policy. This allows the agent to assess Bellman update confidence at the state level. Notably, the update mechanism is algorithm-agnostic, making it a broadly applicable enhancement that can be integrated with various offline RL algorithms.Another key contribution of our approach is demonstrating that state-adaptive regularization mitigates the excessive pessimism of fixed regularization, enabling RL agents to better exploit Bellman updates. Given the heterogeneous quality distribution in offline datasets, this state-wise adaptation offers a promising direction for improving offline RL performance.
> Although our paper does not provide a rigorous theoretical guarantee for the proposed method, given that the regularization encourages the learned policy to mimic dataset actions, the performance lower bound of our method can be approximately ensured by the behavior policy. Furthermore, as our method solely modifies the regularization coefficient, it does not compromise the convergence guarantees of base algorithms such as CQL under standard assumptions.
>
> Thank you for your review again. We hope our response sufficiently addresses your concerns, and we remain available for any further clarifications.
>
> [1] Fu, Justin, Aviral, Kumar, Ofir, Nachum, George, Tucker, Sergey, Levine. "D4rl: Datasets for deep data-driven reinforcement learning". _arXiv preprint arXiv:2004.07219_. (2020).

---

### Official Review · Reviewer_i2Gp · 2025-03-14

**Overall Recommendation:** 4

**Summary:**

This paper proposes to learn a state-adaptive function to dynamically judge how reliable the Bellman update is. This method starts with CQL and transforms the hyperparameter used to modulate how much the value regularizer (i.e. the term responsible for making the value function more conservative) into a parameterized function which relaxes constraints around areas near the dataset and raises them in areas with high uncertainty. The constraint is only applied to state-action pairs in a sub-set of the full dataset to counter issues with low-value data. They apply this value constraint method to policy constraint methods. Finally, they compare empirically to TD3+BC (policy constraint method) and CQL (value constraint method) and show improvement.

**Claims And Evidence:**

The main claim of the paper is that a state-dependent regularizer coefficient in CQL will outperform a fixed parameter. This claim is supported through a series of empirical comparisons between their method and a fixed parameter. While some work needs to go into strengthening their statistical claims (see below), they mostly provide compelling evidence for their approach.

**Essential References Not Discussed:**

N/A

**Experimental Designs Or Analyses:**

W-1. While the design of the experiments are good in theory, I have noticed some odd patterns in the results presented in table 1. Typically a bold-faced result means it is the best performing method w/ statistical significance. But there are several tasks which have overlapping confidence intervals between bolded and non-bolded methods (for instance TD3+BC hopper-expert-v2). While overlapping CIs don't always suggest non-statistical significance, without more details the reader is left to assume these are the standard error, meaning a lack of statistical significance in many of the results of Table 1. This is compounded in Table 2 where baselines don't have confidence intervals reported.

There are several papers you can find that discuss these issues. Here are a few:
- [Deep Reinforcement Learning that Matters](https://ojs.aaai.org/index.php/AAAI/article/view/11694)
- [Empirical Design in Reinforcement Learning](https://www.jmlr.org/papers/v25/23-0183.html)

I would recommend re-thinking how you bold results in your tables, and make sure results you are highlighting are statistically significant. Just having one mean larger than another does not prove it is performing better.

## Hyperparameters

You discuss in detail the hyperparameters added by your approach, but neglect the standard hyperparameter choices. It is possible I missed these, but this should be discussed somewhere in the main text, or at least expanded on in the appendix.

**Methods And Evaluation Criteria:**

Making a parameter state-dependent is generally a great way to make a method more flexible and easier to use. The regularization parameter in CQL is a good candidate for this process. The criteria used for the optimization is reasonable and laid out in an intuitive manner. Overall, I believe the method proposed does a good job at improving other approaches with a fixed coefficient.

**Other Comments Or Suggestions:**

- The phrasing for line 025 in the abstract is a bit odd. The sentence "On the one hand,...". This sentence just lists the new things done by the paper, not an either or (which the phrasing suggests).
- Line 133 second column, you reference equation 4 and 5. I think you meant to reference 3 and 4.
- $\mu$ and $\sigma$ are not defined in the context of equation 6. I believe you mean the mean and standard deviation, as using the initial state distribution here doesn't make sense.

**Other Strengths And Weaknesses:**

This paper is relatively strong. There are some issues, such as the lack of confidence intervals in table 2, which make the paper less appealing. But if the issues in W-1 could be solved (which I believe should be straightforward), then I think this paper is ready to be accepted.

**Questions For Authors:**

N/A

**Relation To Broader Scientific Literature:**

This approach very clearly fits into the literature by making the regularization parameter of CQL a state-dependent function.

**Theoretical Claims:**

N/A

---

> ### Author Rebuttal · Authors · 2025-04-01
>
> Thanks you for the high praise and the comprehensive review of our paper. We appreciate the questions you raised and are committed to delivering a comprehensive response to address the issues.
>
> **Q1: Lack of statistical significance**
>
> **A1:** To ensure statistical rigor, we have now included 95% confidence intervals (CIs) for all domain tasks in Tables 1 and 2, consistent with prior works.  The revised tables are as follows:
>
> **Table 1 (offline performance)**
> |Dataset |TD3+BC | TD3+BC(SA)| CQL |CQL(SA) | Base| Ours |
> |:----------:|:----------:|:----------:|:----------:|:----------:|:----------:|:----------:|
> |...|...|...|...|...|...|...|
> |locomotion total |1000.8 | **1116.7**|1030.4 |**1139.1** |1015.6| **1128.0**|
> |95% CIs|917.9~1083.7|1096.2~1137.3|990.4~1070.1|1111~1167.3|937.5~1078.6|1093.2~1162.8|
> |...| ...|...|...|...|...|...|
> |antmaze total | 131.8|**276.0** |294.1|**406.8**| 213.0| **341.4**|
> |95% CIs | 78.2~185.5|246.5~305.5| 230.9~357.3|334.9~478.7| 130.1~295.8|273.7~409.1|
>
> **Table 2 (offline-to-online performance)**
> |Dataset |IQL |SPOT|FamCQL |TD3+BC |CQL|TD3+BC(SA)|CQL(SA)
> |:---:| :---:|:---:|:---:|:---:|:---:|:---:|:---:|
> |...| ...|...|...|...|...|...|...|
> |locomotion total | 1057.4|  1107.1| 1178.3|1064.5 | 1069.4| 1218.6| **1278.4**|
> |95% CIs| 981.5~1133.2|1093.1~1121.4|1165.3~1191.5|1039.8~1089.2|1058.9~1080.1|1165.4~1248.9|1254.9~1303.5|
> |...| ...|...|...|...|...|...|...|
> |antmaze total |363.2 |  458.1| - |76.8 |405.5 | 389.2|**494.3** |
> |95% CIs |302.8~423.7 | 384.0~534.0| - |79.5~146.1 |327.9~483.1 |316.1~462.4|415.6~573.4|
>
> These results confirm that our method achieves statistically significant performance improvements across different algorithms and task domains.
>
> **Q2: About hyperparameters**
>
> **A2:** The hyperparameters used in our implementation are provided in Appendix D. For fair comparisons, we adopt the same hyperparameter configurations for CQL and TD3+BC as in the CORL benchmark [1].  Additionally, for the distribution-aware threshold, we set $n_{start}$ to 1 and $n_{end}$ primarily to 3 (1.5 for expert datasets to better mimic high-quality actions). We will include a detailed listing of these parameters in the revised version for greater clarity.
>
> We also deeply appreciate your constructive suggestions, including pointing out typographical errors, which we have now corrected.  Thank you again for your thoughtful review and for helping us improve our work.  We hope our responses sufficiently address your concerns, and we remain open to any further questions or clarifications.
>
> [1] Tarasov, Denis, Alexander, Nikulin, Dmitry, Akimov, Vladislav, Kurenkov, Sergey, Kolesnikov. "CORL: Research-oriented deep offline reinforcement learning library". _Advances in Neural Information Processing Systems_ 36. (2023): 30997–31020.

---

### Decision · Program_Chairs · 2025-05-01

**Decision:**

Accept (poster)

**Comment:**

After careful evaluation of the reviews, author rebuttals, and the paper itself, I recommend accepting this submission.  This paper presents a novel method for addressing a key challenge in offline reinforcement learning (RL): balancing between extrapolation errors and behavior cloning in value estimation through state-adaptive regularization. Rather than relying on a global, fixed regularization coefficient, the authors propose a state-adaptive mechanism that dynamically adjusts regularization strength at the state level, informed by the reliability of Bellman updates. Additionally, the paper introduces selective regularization over a subset of high-quality actions, mitigating the degradation caused by low-quality data. The method is unified across both value regularization (e.g., CQL) and explicit policy constraint approaches, and achieves significant improvements in both offline and offline-to-online settings on the D4RL benchmark. The inclusion of ablation studies, and comparisons with recent baselines like FamO2O further bolster the experimental robustness.


There are two limitations the authors could further discuss or address in the final revision. 1. Limited theoretical guarantees: While the paper provides an intuitive and empirical justification for state-adaptive regularization, it lacks a formal theoretical analysis or convergence guarantees under the proposed method. 2. Evaluation beyond continuous control: The experiments are confined to continuous control benchmarks (MuJoCo, AntMaze); evaluating other domains (e.g., discrete control) would further validate generality and make the paper more impactful.